# Role of Non-Coding RNAs in Colorectal Cancer: Focus on Long Non-Coding RNAs

**DOI:** 10.3390/ijms232113431

**Published:** 2022-11-03

**Authors:** Matteo Lulli, Cristina Napoli, Ida Landini, Enrico Mini, Andrea Lapucci

**Affiliations:** 1Department of Experimental and Clinical Biomedical Sciences “Mario Serio”, Section of General Pathology, University of Florence, 50134 Florence, Italy; 2Department of Health Sciences, Section of Clinical Pharmacology and Oncology, University of Florence, 50139 Florence, Italy

**Keywords:** long non-coding RNAs, colorectal cancer, gene regulation, diagnosis, chemoresistance, predictive, prognosis, therapeutic targets

## Abstract

**Simple Summary:**

Non-coding RNAs are a type of RNA that is not translated into proteins. They play important roles in several cellular processes, mainly in the regulation of genetic information transfer from DNA to proteins. Mutations or imbalances in the non-coding RNA repertoire are involved in the development of many human diseases, including cancer. Here, we provide an overview of the role of long non-coding RNAs in the initiation and progression of colorectal cancer as well as in the development of drug resistance and demonstrate their relevance as predictive molecular biomarkers as well as novel potential therapeutic targets.

**Abstract:**

Colorectal cancer is one of the most common causes of cancer-related deaths worldwide. Despite the advances in the knowledge of pathogenetic molecular mechanisms and the implementation of more effective drug treatments in recent years, the overall survival rate of patients remains unsatisfactory. The high death rate is mainly due to metastasis of cancer in about half of the cancer patients and the emergence of drug-resistant populations of cancer cells. Improved understanding of cancer molecular biology has highlighted the role of non-coding RNAs (ncRNAs) in colorectal cancer development and evolution. ncRNAs regulate gene expression through various mechanisms, including epigenetic modifications and interactions of long non-coding RNAs (lncRNAs) with both microRNAs (miRNAs) and proteins, and through the action of lncRNAs as miRNA precursors or pseudogenes. LncRNAs can also be detected in the blood and circulating ncRNAs have become a new source of non-invasive cancer biomarkers for the diagnosis and prognosis of colorectal cancer, as well as for predicting the response to drug therapy. In this review, we focus on the role of lncRNAs in colorectal cancer development, progression, and chemoresistance, and as possible therapeutic targets.

## 1. Introduction

Colorectal cancer (CRC) is one of the third most frequent and lethal cancer worldwide and the second leading cause of cancer-related deaths globally [1]. Most patients have metastases at diagnosis (approximately 20%) or develop them later because of the natural history of the disease (approximately 25%) [2]. Despite the availability of both cytotoxic chemotherapeutic and targeted agents, administered based on the knowledge of some tumor biomarkers, drug treatment response needs improvement because both intrinsic and acquired resistance mechanisms are responsible for treatment failure. Therefore, there is a clinical need to improve the understanding of the biological processes in CRC that are responsible for gene deregulation, heterogeneity, and escape during immunosurveillance and tumor growth control by drug treatment; thus, revealing the role of novel pathogenic determinants and their application as biomarkers for disease characterization, prediction of therapeutic drug response, and prognosis, is crucial [3].

Along with somatic gene mutations and altered gene expression profiles, regulatory non-coding RNAs (ncRNAs) and other epigenomic determinants have been identified as potential biomarkers of CRC initiation, progression, prognosis, and drug treatment response [4].

In this review, we summarize the information on the most well-characterized long non-coding RNAs (lncRNAs) involved in CRC initiation and progression and examine the development of drug resistance and the lncRNAs that can act as potential targets of new drugs for this disease.

## 2. The Non-Coding RNAs and Their Deregulation in CRC

RNA, except transfer and ribosomal RNA, has historically been considered an intermediary between DNA and proteins, restricting its main function to coding. However, the discovery of non-coding-dependent regulatory effects of a plethora of different RNA species has revealed an intricate layer of complexity in the regulation of gene expression, indicating the importance of ncRNAs in this process. Within the ncRNA family, microRNAs (miRNAs), circular RNAs (circRNAs), and lncRNAs play pivotal regulatory roles [5]. The discovery of mutations in the genes encoding ncRNAs involved in cancer phenotypes has changed the perspective of molecular genetics and cancer pathobiology. ncRNAs participate in a wide range of cellular pathways, such as proliferation, differentiation, migration, angiogenesis, and apoptosis [6], which confirms that ncRNAs can also act as oncogenes or tumor suppressors in tumor onset and progression [7].

miRNAs are 18–25 nucleotides long and evolutionarily conserved as single-stranded RNAs [8]. They are processed from larger precursors through sequential cleavages by two RNase-III-like enzymes: Drosha (in the nucleus) and Dicer (in the cytoplasm). By interacting with the protein Ago2, one strand of the resulting duplex can associate with an RNA-induced silencing complex (RISC). In most cases, these miRNA-RISC complexes target specific mRNA by binding to their 3′ untranslated regions (UTRs), which may lead to translational repression or cleavage of the mRNAs [9]. Deregulation of miRNA expression during CRC onset and progression has been extensively described, along with a series of up- or down-regulated miRNAs that can predict the response to neoadjuvant therapy [10]. Among these, four deregulated miRNAs have been identified as biomarkers: *miRNA-145,* which targets *TRIAP1*, *EGFR*, and *BAK1; miRNA-223* and *miRNA-1246*, both of which target *KRAS*; and *miRNA-622*, which targets *TP53, KRAS, HIF1A, VEGFA, EGFR*, and *MKI67* [10]. Recently, we discovered that *miRNA-548I1* is up-regulated in a cohort of poor-prognosis patients with stage III CRC undergoing adjuvant chemotherapy [11]. The function of this miRNA had not been characterized before our study, and it had been not reported to be associated with any type of cancer. Using TargetScan bioinformatic analysis, 22 putative targets of *miRNA-548I1* were identified. Interestingly, some of them are deregulated in CRC (e.g., microtubule-associated protein RP/EB family member 1 and angiotensin II receptor type 1) [11].

CircRNAs are another class of regulatory RNAs, which can be divided into non-coding and coding circRNAs [12]. Their molecular structure is characterized by a covalent closed-loop structure without a 5′ cap and/or 3′ poly A tail [13]. Based on their different cyclizations, circRNAs can be further divided into three other subclasses: spliceosome-dependent cable tail patching circRNAs, cis-acting element-promoted circRNAs, and RNA-binding protein-regulated circRNAs [14,15]. CircRNAs display cell-type, tissue-type, and developmental stage-specific expression patterns in the eukaryotic transcriptome; thus, they are involved in gene expression regulation by sponging specific miRNAs, thereby preventing their binding to target mRNAs, or by interacting with RNA-binding proteins and influencing mRNA splicing and stability [16]. Many studies have demonstrated the close association of circRNAs with several physiological and pathological pathways of cancer, including growth, differentiation, metastasis, and invasion [17], and established the pivotal role of circRNAs in CRC patients with lung metastasis [18]. *CircRNA YAP1* has been shown to inhibit the proliferation and invasion of gastric cancer cells [19], and *circITGA7* promotes tumor growth and metastasis in CRC tissues and cell lines by down-regulating its gene and the linear host gene *ITGA7* [20]. Furthermore, it has been recently reported that an open reading frame with translational function is present in *circRNA PPP1R12A;* the proteins that it encodes affect the proliferation, migration, and invasion of CRC cells by activating the Hippo-YAP signaling pathway [21].

Bhuyan et al. focused on differentially expressed circRNAs to investigate their role in CRC onset [22]. They analyzed 122 differentially expressed circRNAs in total; among them, 19 are highly up-regulated and 3 are highly down-regulated in CRC. All these circRNAs are transcribed by 10 coding genes. The circRNAs *hsa_circ_0024107* derived from the *MMP1* gene and *hsa_circ_0044518* derived from the *COL1A1* gene are over-expressed in CRC patients, and they both act as sponges for many miRNAs; additionally, their up-regulation is closely related to the development and progression of CRC [22]. Among the down-regulated genes, solute carrier family 26 member 2 (*SLC26A2*) produces the circRNA *has_circ_0074494*, which is down-regulated in CRC cancer cell lines compared with that in normal colon epithelial cell lines, and actin gamma 2 smooth muscle (*ACTG2*) produces *hsa_circ_0055267* and *hsa_circ_0055266* that inhibit cell growth and reduce cell viability of small intestinal neuroendocrine tumor cells [23].

The last class of ncRNAs extensively reviewed here is lncRNAs. They are defined as transcripts longer than 200 nucleotides that generally lack coding potential and can be processed like mRNAs, i.e., spliced or polyadenylated. Human lncRNAs can be divided into two main classes: intergenic, genic, or intragenic lncRNAs. The lncRNA family comprises long intergenic ncRNAs (lincRNAs), macro and very long intergenic ncRNAs (vlincRNAs), enhancer- or promoter-associated RNAs (eRNAs or paRNAs), natural antisense transcripts, and primary RNA polymerase II transcripts [24]. LncRNAs regulate gene expression at multiple levels, mainly interacting with DNA, RNA, and proteins [25]. Relative to the interaction with DNA, several lncRNA-dependent epigenetic roles have been revealed. LncRNAs possess a direct regulatory potential on chromatin architecture and related gene expression, by virtue of their negative charge which can neutralize the positively charged histone tails, thus inducing chromatin de-compaction [25]. Additionally, lncRNAs affect accessibility to chromatin through the formation of RNA-DNA hybrid structures, such as triple helices (RNA-DNA-DNA) or R-loops, which are involved in gene expression control, genome stability, and DNA repair regulation [26]. LncRNAs mediate the recruitment and the activity of specific DNA regions, or function as decoys of chromatin modifiers, such as histone deacetylases. Furthermore, lncRNAs play inducing or repressing roles in transcription, by interfering with transcription machinery or modulating the activity of enhancers and promoters. Considering the functional roles of lncRNAs which depend on their interaction with RNA and proteins, numerous modes of lncRNA functioning as post-transcriptional, translational, and post-translational regulators have been extensively described. By virtue of their RNA sequence and/or structure, lncRNAs bind to and sequester RNA-binding proteins, thereby regulating mRNA splicing, localization, turnover, and translational rates. In addition, lncRNAs can directly base pair with other coding and non-coding RNAs. Relative to this last case, it is well known that lncRNAs can bear miRNA-complementary target sites, thus acting as competitive endogenous RNAs through sponging these miRNAs, and, in turn, reducing their availability to target mRNAs [27]. Deregulation of lncRNA-dependent regulative mechanisms has been described in CRC initiation, progression, and acquisition of drug resistance (Figure 1), and paradigmatic examples of these alterations are reviewed below.

## 3. LncRNAs in Initiation and Progression of CRC

CRC develops from normal mucosal epithelium to a benign adenoma and finally progresses to become a malignant tumor. The pathogenesis of CRC is a multi-step process driven by genetic and epigenetic alterations that perturb cellular physiology [28,29]. These alterations initiate an evolutionary process mainly characterized by the acquisition of hallmark capabilities for transforming CRC cells, such as sustaining proliferative signaling, evading growth suppressors, resisting apoptosis, enabling replicative immortality, and activating the epithelial-mesenchymal transition (EMT) program, angiogenesis, invasion, and metastasis. It has become clear in recent years that the alteration of numerous molecular mechanisms driven or coordinated by lncRNAs is directly involved in all of these cancer hallmarks. Considering the role of some lncRNAs in CRC initiation and progression, lncRNAs play a crucial role in the regulation of complex cellular processes, and the deregulation of a lncRNA can determine the phenotypic alteration of several tumor processes simultaneously, such as the induction of proliferation, invasive capacity, and metastasis of tumor cells.

Understanding the involvement of lncRNAs in tumorigenic signaling pathways, such as Wnt/β-catenin, epidermal growth factor receptor (EGFR)/insulin-like growth factor 1 receptor (IGF-1R), KRAS, phosphatidylinositol-3-kinase (PI3K), transforming growth factor-beta (TGF-β), p53, and EMT signaling pathways [30] can provide further insights into CRC pathogenesis.

Growing evidence indicates that the lncRNA regulator of reprogramming (*lnc-ROR*) regulates the progression of various cancers by promoting the proliferation, invasion, migration, and drug resistance of various cancer cells, including lung cancer, hepatocellular carcinoma, breast cancer, and CRC cells [31]. In CRC cell lines, *lnc-ROR* sponges miRNAs that regulate stem cell factors such as *POU class 5 homeobox 1*, *Nanog,* and *SRY-box 2*; it also reduces sensitivity to radiotherapy by deregulating the *p53/miR-145* pathway [32]. Over-expression of *lnc-ROR* is also associated with the EMT pathway activation and metastases in CRC, and it down-regulates the expression of *miR-6833-3p*, thereby inhibiting the apoptosis-related protein SMC4 [33]. Deregulation of lncRNA expression occurs in a tissue- or organ-specific way [34,35,36] and appears to be strongly related to CRC onset and progression [37].

Deregulation of a lncRNA has been observed to be associated with the loss of imprinting of long QT intronic transcript 1 (*LIT1/KCNQ1OT1*) in CRC, indicating its potential as a useful marker for CRC diagnosis [38]. The oncogenic lncRNA HOX antisense intergenic RNA (*HOTAIR*) binds to polycomb repressive complex 2 (*EZH2*) and lysine-specific histone demethylase 1A (*KDM1A*) in the 5′ and 3′ regions and represses the transcription of the homeobox D cluster (*HOXD*) family genes in breast cancer progression and CRC by acting as a scaffold for histones [37]. The high expression of *HOTAIR* is significantly correlated with distant metastasis and poor prognosis in CRC patients [39], and it has not only been confirmed as a negative prognostic factor in primary tumors but also as a circulating biomarker in the blood of CRC patients [40].

Moreover, the lncRNA metastasis-associated lung adenocarcinoma transcript 1 (*MALAT1*), which acts as a predictive biomarker of metastasis in non-small cell lung cancer patients, has been described as a new prognostic marker in CRC patients [41,42]. The lncRNAs *MALAT1*, *CCAT1*, and *PANDAR* are up-regulated in the blood of CRC patients compared with that in healthy controls, suggesting their role as potential biomarkers for CRC prognosis [43].

Recently, Liu et al. [44] identified a competing endogenous RNA (ceRNA) regulatory network by bioinformatics analysis and experimental validation, in which 23 differentially expressed lncRNAs, 7 miRNAs, and 244 mRNAs act as regulatory axes associated with CRC tumorigenesis and prognosis. In the *IGF2-AS/miR-150/IGF2* axis, the over-expression of *miRNA150* down-regulates the expression of the lncRNA *IGF2-AS*, resulting in the over-expression of *IGF2*; however, *IGF2-AS* expression is positively correlated with *IGF2* expression in CRC patients. Another lncRNA associated with the pathogenesis of CRC is plasmacytoma variant translocation 1 (*PVT1*); its up-regulation influences the down-regulation of *miR-16-5p*, which plays a significant role as a tumor suppressor in CRC [45]. The loss of *PVT1* and *miR-16-5p* over-expression has been observed to drastically reduce the tumor volume in a mouse xenograft model. Furthermore, the *PVT1-miR-16-5p/VEGFA/VEGFR1/AKT* axis is directly associated with CRC pathogenesis: *PVT1* up-regulation induces the down-regulation of *miR-16-5p*, thereby reducing the binding of this miRNA to the mRNA *VEGFA* and up-regulating *VEGFA*, affecting VEGFR1 and AKT signaling [45]. Genome-wide association studies have identified a single nucleotide polymorphism at the *PVT1* locus (8q24) that is closely associated with an increased risk of developing CRC [46]. The *PVT1* locus drives the production of four miRNAs: *miRNA-1204, miRNA-1205, miRNA-1206*, and *miRNA-1207-5p* and *-3p*, some of which are important in the tumorigenic onset of CRC and gastric cancer [47,48,49,50].

Among the differentially expressed lncRNAs implicated in CRC progression, *RP11-468E2.5* is less expressed in CRC samples than in paired normal mucosa, whereas its target genes *STAT5A* and *STAT6* transcription factors are up-regulated [51]. Furthermore, the silencing of *RP11-468E2.5* leads to an increase in the expressions of *JAK2, STAT3, STAT5, STAT6, CCND1*, and *Bcl-2* and a decrease in *P21* and *P27* expressions, highlighting the effects of *RP11-468E2.5* down-regulation on the activation of the JAK/STAT signaling pathway, which is driven by the increase of *STAT5* and *STAT6*, promoting cell proliferation and inhibiting apoptosis in CRC [52].

The lncRNA *MIR17HG* is also associated with carcinogenesis and CRC progression [53]. *MIR17HG* induces *NF-κB/RELA* expression by sponging *miR-375*. RELA activates the transcription of *MIR17HG* by binding directly to its promoter region in a positive feedback loop [54]. Among the various miRNAs transcribed by *MIR17HG*, *miR-17-5p* reduces the expression of the tumor suppressor B-cell linker (*BLNK*), providing CRC cells greater capacity to migrate and invade. Furthermore, *MIR17HG* up-regulates *PD-L1* expression, thus representing a potential target for immunotherapy.

The lncRNA *MFI2-AS1* sponges *miR-574-5*, activating the expression of MYC binding protein (*MYCBP*) and thus promoting the proliferation and migration of CRC cells [55].

The lncRNA FEZF1 antisense RNA 1 (*FEZF1-AS1*) is also closely associated with cell proliferation, migration, and invasion in both CRC cell lines and patients. Reduced expression levels of *FEZF1-AS1* inhibit the activation of the EMT pathway and increase the expression levels of orthodenticle homeobox 1 (*OTX1*) [56]; thus, the *FEZF1-AS1/OTX1/EMT* axis is involved in CRC development. In addition, *FEZF1-AS1* positively regulates the expression of *NT5E* by sponging *miR-30a-5p*.

The up-regulation of the lncRNA DNAJC3 divergent transcript (*DNAJC3-DT*) is also strongly associated with CRC progression via modulation of the *miR-214-3p*/*LIVIN* complex. Analysis of the downstream pathway by *DNAJC3-DT* silencing showed that this deregulation increases the expression levels of *miRNA-214-3p* in CRC cell lines. Therefore, the up-regulation of *miRNA-214-3p* inhibits the protein expression of *LIVIN* and suppresses the activation of the NF-κB signaling pathway, which, in turn, inhibits CRC progression [57]. The lncRNA IGFL2 antisense RNA 1 (*IGFL2-AS1*) is involved in angiogenesis and EMT and KRAS signaling. It has recently been shown to be over-expressed in colon adenocarcinoma tissue and CRC cell lines and related to tumor cell proliferation and invasion. *IGFL2-AS1* is an independent tumor marker in colon adenocarcinoma patients. High levels of *IGFL2-AS1* are correlated with worse prognosis and may facilitate cancer progression [58].

## 4. Role of lncRNAs in CRC Drug Resistance

Positive or negative modulation of signaling pathways is the basis for maintaining cellular homeostasis, and alterations in these pathways can cause the development of diseases, such as cancer. LncRNAs are regulators of signaling pathways, and their deregulation may be closely related to the onset of cancer chemoresistance. This section highlights the results of various in vitro and in vivo studies related to the development of CRC chemoresistance associated with the deregulation of lncRNAs and their related signaling pathways. Several studies have been conducted regarding the potential contribution of lncRNAs to the onset of drug resistance in CRC [59]. The first example is the lncRNA KCNQ1 opposite strand/antisense transcript 1 (*KCNQ1OT1*), which is involved in the enhancement of methotrexate resistance. *KCNQ1OT1* regulates the protein phosphatase 1 regulatory inhibitor subunit 1B (PPP1R1B) by sponging *miR-760,* thereby influencing the downstream cAMP responsive element binding protein 1 (CREB1) signaling pathway in CRC cells [60]. In addition, CREB1 is known as a proto-oncogenic transcription factor that modulates the expression of some crucial genes and miRNAs. Furthermore, *KCNQ1OT1* up-regulation was observed to promote resistance to oxaliplatin in CRC cells and in an in vivo model by sponging *miRNA-34a*. This mechanism positively regulates the expression levels of the autophagy-related 4B cysteine peptidase (*ATG4B*) gene, a member of the autophagy protein family, which can enhance the protective autophagy pathway and chemoresistance [61]. Moreover, our group recently identified *KCNQ1OT1* and the coding gene pinin, desmosome-associated protein (*PNN*) as predictive biomarkers of 5-fluorouracil response and outcome in stage III CRC patients. *KCNQ1OT1* over-expression is associated with poor prognosis and the concomitant onset of resistance to 5-fluorouracil. Further studies are warranted to elucidate the signaling pathways influenced by *KCNQ1OT1* activity and their potential correlations with drug resistance in CRC patients [11]. Recently, our group demonstrated that stage II-III CRC patients with high *KCNQ1OT1* expression levels exhibit a shorter disease-free survival (DFS) than those with low *KCNQ1OT1* expression levels. Patients with *KCNQ1OT1* expression values below the identified cut-off in tumor tissues have significantly longer DFS than those with expression levels above the selected cut-off [62].

The lncRNA taurine up-regulated gene 1 (*TUG1*) is also associated with methotrexate resistance through its sponging activity on *miRNA-186*, which increases the expression of cytoplasmic polyadenylation element binding protein 2 (*CPEB2*) in colon cancer cell lines and in triple-negative breast cancer with increased metastasis [63]. In CRC cell lines, TGF-β promotes cell migration by up-regulating lncRNA *TUG1* expression, while its knockdown inhibits migration, invasion, and the EMT pathway in CRC cells in vitro and reduces CRC lung metastasis in vivo [63]. This finding highlights the effects of TGF-β on metastasis via the *TUG1/TWIST1/EMT* signaling pathway in human CRC models [64].

Over-expression of *XIST* and *ROR1* lncRNAs and concomitant down-regulation of *miRNA-30a-5p* also contribute to multidrug resistance in CRC cell lines and tissues [65].

Using RNA-Seq, RT-qPCR, and bioinformatics analysis, Zinovieva et al. identified five lncRNAs (*LINC00973, LINC00941, CASC19, CCAT1*, and *BCAR4*) in a murine CRC xenograft model as well as in HT-29 and HCT-116 CRC cell lines treated with 5-fluorouracil, oxaliplatin, and irinotecan at different concentrations and exposure times [66]. The most frequent changes were associated with *LINC00973*, which was most strongly and consistently increased in CRC cell lines treated with the aforementioned anticancer drugs. Recently, the lncRNA *LINC00973* was characterized as a part of a ceRNA through its sponging activity on *miRNA-7109-3p*; this controls *Siglec-15* expression, a critical immune suppressor, which is highly expressed in human cancer cells [67]. *LINC00973* is also up-regulated in the cetuximab-resistant CRC cell line H508/CR; however, its silencing by a short interfering RNA (siRNA) reduces cell viability, increases apoptosis, and decreases glucose consumption and lactate secretion [68].

*GIHCG*, another important lncRNA, is over-expressed in several CRC cell lines and tissues; it promotes CRC cell proliferation and survival by inhibiting *miR-200b/200a/429* expression and has been associated with resistance to 5-fluorouracil and oxaliplatin in CRC. High *GIHCG* expression is correlated with lymphovascular invasion, lymph node metastasis, and distant metastasis in CRC. Its over-expression also contributes to the resistance to anticancer drugs; patients with high expression levels of *GIHCG* show lower rates of overall survival and progression-free survival [69].

The lncRNA *CACS15* is up-regulated in oxaliplatin-resistant CRC tissues and cells. It contributes to oxaliplatin resistance by positively regulating the expression levels of *ABCC1* by sponging *miR-145* [70]. *CASC15* silencing overcomes oxaliplatin resistance in CRC by regulating the *CASC15/miR-145/ABCC1* axis [71].

A lncRNA closely associated with resistance to 5-fluorouracil in CRC is H19 imprinted maternally expressed transcript (*H19*) through its sponging activity on *miR-194-5p*. This interaction leads to the up-regulation of *SIRT1* and activation of the autophagic pathway, which may play a key role in drug resistance in several cancers, including CRC [72]. SIRT1, a member of the sirtuin family, removes acetyl groups from lysine residues of histones and non-histone proteins, driving gene expression and inducing several signaling pathways, including the autophagy pathway [73]. *H19* also mediates resistance to methotrexate by modulating Wnt/β-catenin signaling in methotrexate-resistant CRC cell lines [74].

The over-expression of the lncRNA small Cajal body-specific RNA 2 (*SCARNA2*) is positively correlated with drug resistance and bad prognosis in CRC patients; down-regulation of *SCARNA2* by RNA silencing was found to restore the drug sensitivity *in vitro*.

*SCARNA2* promotes drug resistance by suppressing the *miR-342-3p* target sequence to modulate *EGFR* and B-cell lymphoma 2 (*Bcl2*) expression in CRC cells, thus entering the *miR-342-3p-EGFR/BCL2* pathway [75].

The lncRNA X-inactive specific transcript (*XIST*) promotes doxorubicin resistance by sponging *miRNA-124* and up-regulates serum and glucocorticoid-inducible kinase 1 (*SGK1*) in CRC cells and tissues [76]. Knockdown of *XIST* reverts resistance to doxorubicin, reduces *ABCB1* and *GSTP1* expression levels, and enhances apoptosis. Over-expression of *miRNA-124* suppresses *XIST*-mediated effects, restoring the sensitivity of CRC-doxorubicin-resistant cell lines [76]. Moreover, *XIST* is involved in 5-fluorouracil resistance by up-regulating thymidylate synthase (*TYMS*) expression levels in CRC cells [77].

Another lncRNA associated with resistance to chemotherapeutic agents in CRC is the lncRNA BRAF-activated non-protein coding RNA (*BANCR*), which is over-expressed in CRC patients. *BANCR* causes doxorubicin resistance by modulating the *miRNA-203*/chromosome segregation 1 like (*CSE1L*) complex in tumor cells [78].

Wang et al. showed that down-regulation of the lncRNA maternally expressed gene 3 (*MEG3*) is closely associated with oxaliplatin resistance through the regulation of the *miR-141*/programmed cell death 4 (*PDCD4*) complex [79]. *PDCD4* is a well-known tumor suppressor gene involved in many types of cancers, including CRC [80].

The lncRNA urothelial carcinoma-associated-1 RNA (*UCA1*) is associated with resistance to cetuximab. An increase in *UCA1* expression levels in cetuximab-resistant CRC patients was negatively correlated with survival time. Moreover, circulating *UCA1* promotes metastasis through the *miR-143/MYO6* axis. *UCA1* also causes resistance to 5-fluororacil by sponging *miRNA-204-5p*; a *UCA1-miR-204-5p-CREB1/BCL2/RAB22A* regulatory pathway is involved in 5-fluorouracil-resistance in CRC patients [81,82,83].

The lncRNA small nucleolar RNA host gene 15 (*SNHG15*) is also associated with resistance to 5-fluorouracil in CRC. By analyzing RNA-seq data of tumor tissue and paired normal mucosa obtained from 456 CRC patients, it was found that over-expression of *SNHG15* in tumors was highly correlated with poor patient outcomes. Higher *SNHG15* expression in tumors with high levels of *MYC* expression and direct modulation of *SNHG15* transcription by the oncogene *MYC* were also observed. The inhibition of *SNHG15* causes changes in multiple relevant genes implicated in cancer progression, including *MYC*, *NRAS*, *BAG3*, and *ERBB3*. Many of these genes are functionally related to apoptosis-induced factor (AIF), a protein that specifically interacts with *SNHG15*, suggesting that *SNHG15* acts, at least partly, by regulating AIF activity and promotes cell proliferation, invasion, and drug resistance in CRC [84].

Another lncRNA associated with oxaliplatin resistance and promotion of metastasis is colorectal neoplasia differentially expressed (*CRNDE*), which is over-expressed in CRC patients and cell lines. *CRNDE* acts as a ceRNA, sponging *miR-136* and driving the reactivation of its target E2F transcription factor 1 (*E2F1*) [85].

*HOTAIR* has been investigated as a putative marker involved in 5-fluorouracil resistance by promoting *TYMS* expression. *HOTAIR* overexpression inhibits 5-fluorouracil-induced cytotoxicity in CRC cell lines. High levels of *HOTAIR* are associated with poor response to 5-fluorouracil-based chemotherapy in CRC patients. The mechanism of resistance is closely related to the role of *HOTAIR* in the negative regulation of *miRNA-218* expression and activation of the NF-κB pathway in CRC patients [86].

Coffey et al. identified that the lncRNA *mir-100-let-7a-2-mir-125b-1* cluster host gene (*MIR100HG*) and two associated miRNAs, *miR-100* and *miR-125b*, are over-expressed in cetuximab-resistant CRC patients and head and neck squamous cell cancer cell lines [87]. *miRNA-100* and *miRNA-125b* can repress some Wnt/β-catenin negative regulators, resulting in increased Wnt signaling, whereas Wnt inhibition in cetuximab-resistant cells restores cetuximab responsiveness. The mechanism of cetuximab resistance represents a double-negative feedback loop between *MIR100HG* and the transcription factor GATA6; although GATA6 represses *MIR100HG*, this is hampered by its targeting of *miR-125b* [87].

Peng et al. observed that POU class 5 Homeobox-1 pseudogene 4 (*POU5F1P4*) down-regulation reduces the sensitivity of metastatic CRC cells to cetuximab and could be a potential new treatment for metastatic CRC [88].

Over-expression of the lncRNA prostate cancer-associated Transcript-1 (*PCAT-1*) promotes prostate cancer cell proliferation through *MYC* and is closely associated with poor prognosis in CRC patients. Silencing *PCAT-1* in CRC cells suppresses cell motility and invasiveness and increases the response to 5-fluorouracil treatment [89]. Another lncRNA closely related to 5-fluorouracil resistance in CRC is HOXA transcript antisense RNA myeloid-specific 1 (*HOTAIRM1*); it is present in lower levels in 5-fluorouracil-resistant CRC tissues and cell lines (HCT116 and SW480). *HOTAIRM1* can induce drug resistance, together with B-cell translocation gene 3 (*BTG3*), which is a target of *miRNA-17-5p*. *BTG3* is a p53 target that binds to the transcription factor *E2F1*, inhibiting its expression. *BTG3* suppresses AKT activity, which is frequently deregulated in many cancers [90]. Thus, the over-expression of *miRNA-17-5p* causes down-regulation of *HOTAIRM1* and *BTG3*, resulting in increased resistance to 5-fluorouracil in CRC cells [91].

The main lncRNAs involved in the chemoresistance of CRC are summarized in Table 1.

## 5. LncRNAs as Therapeutic Targets

There is a growing interest in the use of lncRNAs as modulators of drug action to overcome drug resistance. In a recent review, Wang et al. analyzed the possible development of new targeted drugs, with a focus on modulating lncRNAs that can be used as targets in a new therapeutic approach to reverse the resistant phenotypes [92]. LncRNAs can be targeted by multiple approaches, as briefly described below.

First, lncRNA-mediated post-transcriptional RNA regulation pathways can be altered by using siRNAs, miRNAs, and Dicer and Argonaute protein-dependent cleavage pathways that are central to RNA silencing pathways [93]. In a phase I clinical trial (NCT03087591) based on the use of siRNA-transfected peripheral blood mononuclear cells (PBMCs) APN401, the side effects and effective dose of APN401 in the treatment of patients with CRC, pancreatic cancer, and other solid tumors were evaluated. APN401 may inhibit the growth of cancer cells by blocking some of the key enzymes involved in carcinogenesis. In this study, autologous PBMCs were transfected ex vivo with a construct encoding siRNA that targeted Cbl proto-oncogene B (*Cbl-B*). Silencing the *Cbl-B* gene increases cytokine production and proliferation and activation of the immune system, leading to cancer cell death. In addition, *Cbl-B* is a negative regulator of the immune system and is highly mutated in several cancer types; *Cbl-B* expression levels are inversely correlated with the activation of T-lymphocytes and tumor cell death [94]. Recently, Hu et al. reviewed the state of the art of using siRNAs in clinical practice; only two approved siRNA-based drugs are currently in use: ONPATTRO^®^ (Patisiran, ALN-TTR02) for the treatment of TTR-mediated amyloidosis and GIVLAARI^™^ (Givosiran, ALN-AS1) for the treatment of acute hepatic porphyrias [95]. Many other siRNA-based drugs for various diseases are currently in different phases of clinical trials.

Therapeutic strategies based on new-generation antisense oligonucleotides (ASOs) (e.g., locked nucleic acids, LNAs) can be used to target and repress the expression of specific lncRNAs. LNAs are single-stranded oligonucleotide that contains a DNA stretch flanked by locked nucleic acid nucleotides. The mechanism of action of these antisense oligonucleotides includes the activation of the RNase H-mediated degradation pathway of the target sequence. ASO-mediated silencing of *MALAT1* function has been found to reduce metastasis in lung cancer cells and a murine xenograft model [96]. Inhibiting *MALAT1* by specific LNAs in human endothelial cells and in a mouse model caused a functional reduction in the recovery of blood flow and capillary density after hind limb ischemia [97].

A recent review by Nedaeinia et al. analyzed the effects of LNA against *miRNA-21* and its pharmacological inhibitory effects as a potential therapeutic option in the treatment of CRC, providing an overview of preclinical and clinical studies related to the targeting of key dysregulated RNAs in CRC [98]. These therapeutic strategies could be translated not only for targeting miRNAs but also for modulating lncRNA expression in CRC patients.

The epigenetic regulatory pathways of lncRNAs can be altered by the modulation of lncRNA transcription using steric blockade of the promoter or by genome-editing techniques (knockdown) such as the clustered regularly interspaced short palindromic repeats (CRISPR-Cas) system. However, no clinical studies on this exist; studies have been conducted only on cellular and animal models of CRC. Yu et al. recently demonstrated that the knockout of *CCAT2* by CRISPR-Cas increased *miRNA-145* expression levels and decreased *miRNA-21* expression in the HCT-116 cell line; this genetic knockout produced a decrease in the proliferation and differentiation of cancer cells [99]. Using the CRISP-Cas system, Bester et al. recently identified the lncRNA *GAS6-As2* as the main regulator of the GAS6/TAM pathway in a panel of tumor cell lines; over-activation of this pathway leads to resistance in multiple cancer types, including CRC [100].

The approach based on the CRISPR-Cas system represents a cutting-edge tool for fundamental research and could be a new therapeutic approach in cancer treatment. In addition, developments in nanotechnology materials, safe viral delivery systems, and the synthesis of next-generation liposomes, polymers, nanoparticles, and peptides will increase the possibility of using the CRISPR-Cas system in cancer cells of patients [101].

## 6. Conclusions

In this review, we summarized the role of lncRNAs and their deregulation in CRC as predictive biomarkers for diagnosis, prognosis, and chemoresistance. We also highlighted the possible therapeutic strategies that can target lncRNAs, which will increase the sensitivity of CRC cells to conventional chemotherapy drugs and targeted therapy.

The role of lncRNAs in CRC development, progression, and acquisition of chemoresistance has been confirmed by many researchers. The mechanisms by which lncRNAs act can be categorized into four groups: epigenetic regulation, sponging of miRNAs, interaction with regulatory proteins, and modulation of signaling pathways. Emerging evidence has demonstrated that lncRNAs play crucial roles in CRC hallmarks through various molecular mechanisms and therefore have the potential to be utilized for new targeted drug development [102]. However, identification and validation of lncRNAs that can avoid non-specific targets and provide safe delivery systems are currently in development and early clinical trial phases but remain challenging.

With the increasing knowledge of lncRNA deregulation and its implications in CRC, it is likely that some of these lncRNAs will benefit biomarker-oriented precision medicine approaches. Furthermore, current knowledge regarding lncRNAs provides the basis for further development of medical applications of lncRNAs as a CRC disease index, a prognostic factor of chemotherapy response, a disease-outcome marker of CRC patients, and a therapeutic target.

As discussed above, lncRNAs can interact with ncRNAs, DNA, mRNA, and proteins as regulators of normal cellular physiology. Deregulation of lncRNAs can occur during the onset and progression of CRC and can form the basis for therapeutic approaches. Although a wide-ranging analysis and integrated approach investigating the association of lncRNAs with ncRNAs, proteins, and genes will be required to successfully develop effective therapies, the combination of the lncRNAs mentioned in this review in future therapeutic tools can provide new treatment opportunities in the management of CRC patients.

## Figures and Tables

**Figure 1 ijms-23-13431-f001:**
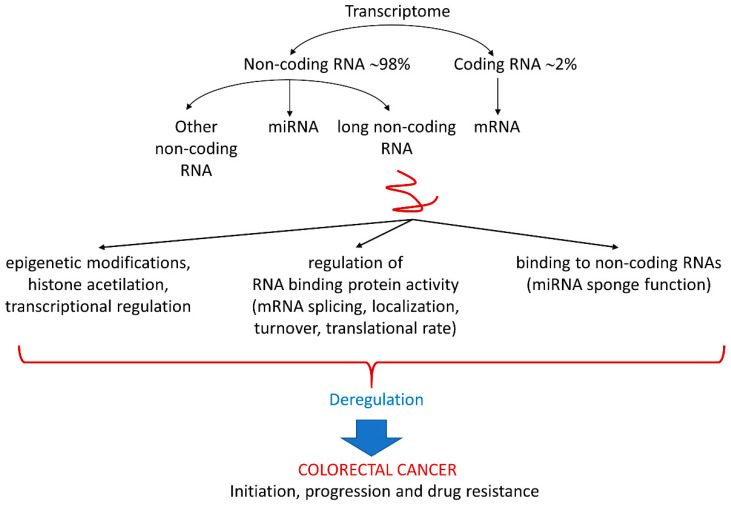
Alteration of lncRNAs molecular mechanisms in CRC initiation, progression, and drug resistance.

**Table 1 ijms-23-13431-t001:** LncRNAs involved in CRC Chemoresistance.

Official Symbol	Official Full Name	Chromosomal Localization	Expression Levels in CRC	Drugs Substrate
*KCNQ1OT1*	KCNQ1 Opposite Strand/Antisense Transcript-1	11p15.5	Up-regulated	Methotrexate resistance 5-FU resistance
*TUG1*	Taurine Up-Regulated 1	22q12.2	Up-regulated	Methotrexate resistance
*XIST*	X Inactive Specific Transcript	Xq13.2	Up-regulated	Multidrug resistance Doxorubicin resistance 5-FU resistance
*LINC00973*	Long Intergenic non-protein Coding RNA 973	3q12.1	Up-regulated	Cetuximab resistance
*GIHCG*	GIHCG Inhibitor of miR-200b/200a/429 Expression	12q14.1	Up-regulated	5-FU and Oxaliplatin resistance
*CASC15*	Cancer Susceptibility-15	6p22.3	Up-regulated	Oxaliplatin resistance
*H19*	H19 Imprinted Maternally Expressed Transcript	11p15.5	Up-regulated	Methotrexate resistance 5-FU resistance
*SCARNA2*	Small Cajal Body-Specific RNA-2	1p13.3	Up-regulated	Chemoresistance
*BANCR*	BRAF-Activated non-protein Coding RNA	9q21.11-q21.12	Up-regulated	Adriamycin resistance
*UCA1*	Urothelial Cancer Associated-1	19p13.12	Up-regulated	Cetuximab resistance 5-FU resistance
*SNHG15*	Small Nucleolar RNA Host Gene 15	7p13	Up-regulated	5-FU resistance
*CRNDE*	Colorectal Neoplasia Differentially Expressed	16q12.2	Up-regulated	Oxaliplatin resistance
*HOTAIR*	HOX Transcript Antisense RNA	12q13.13	Up-regulated	5-FU resistance
*MIR100HG*	mir-100-let-7a-2-mir-125b-1 Cluster Host Gene	11q24.1	Up-regulated	Cetuximab resistance
*LINC00473*	Long Intergenic non-protein Coding RNA 473	6q27	Up-regulated	Taxol resistance
*PCAT1*	Prostate Cancer Associated Transcript-1	8q24.21	Up-regulated	5-FU resistance
*MEG3*	Maternally Expressed-3	14q32.2	Down-regulated	Oxaliplatin resistance
*POU5F1P4*	POU Class-5 Homeobox-1 Pseudogene-4	1q22	Down-regulated	Cetuximab resistance
*HOTAIRM1*	HOXA Transcript Antisense RNA Myeloid-Specific 1	7p15.2	Down-regulated	5-FU resistant

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
