# Peer review of "Role of Non-Coding RNAs in Colorectal Cancer: Focus on Long Non-Coding RNAs"

_ijms, 2022, doi:10.3390/ijms232113431_

Round 1

Reviewer 1 Report

The topic of this manuscript is of great interest to the audience. However, the typos and grammar errors in the manuscript led to bad reading experience. The authors just listed the results from different studies instead of organizing/integrating them in a more efficient way. Thus, I would suggest rejection for the current version manuscript due to its defect in both language and structure.

1.     Typo in title: Role of non-coding “RNAs”

2.     The last row of the second paragraph in page 2: tumor suppressortumor suppressors

3.     I feel it is more common to use “regulatory RNAs” rather than “regulative RNAs”

4.     The fourth row from the bottom of page 2: remove one “in”

5.     The third row of the second paragraph in page 3: polyadenilatedpolyadenylated

6.     Figure 1: Hystone Acetilationhistone acetylation; bindings proteinbinding proteins

7.     The nineth row from the bottom of page 4: axesaxis

8.     The third row from the bottom of page 4: I1It

9.     Use same font size throughout the main text.

10.  Reference (58) in page 5: keep the format of reference consistent throughout the manuscript

11.  The third and the ninth rows of the third paragraph in page 9: LANsLNAs

12.  The third, fifth, nineth, and thirteen rows in page 10: CRISP-CasCRISPR-Cas

Author Response

We thank the Reviewer for his/her suggestions, which allow us to improve the quality of our manuscript. The manuscript has been extensively edited and reviewed by Editage, a professional English language editing and publication support service. The certificate of editing is attached. This editing solve point-by-point the typos and English term mistakes indicated by the Reviewer.

According to the Reviewer’s suggestion, we modified the manuscript as follow: section 1 relates to introduction of the issue; section 2 relates to non-coding RNAs presentation, description and a new deeper description of molecular mechanisms, with a particoular focus on CRC; section 3, which includes a completely new introduction, relates to roles of lncRNAs in initiation and progression of CRC; section 4 describes various in vitro and in vivo studies related to the development of CRC chemoresistance associated with the deregulation of lncRNAs; section 5 states the state of the art of lncRNAs as therapeutic targets in CRC; section 6 includes conclusive perspectives. In addition, we improved figure 1 and legend accordingly to new descriptions of molecular mechanisms.

Reviewer 2 Report

This review describes recent advances of functional roles and clinical applications of lncRNAs in CRC. However, authors may provide more detailed information about molecular mechanisms of lncRNAs. There are many grammatical errors throughout the text. Some of English terms and writing style need to be improved. Editing the manuscript by an English speaker is highly recommended. The content of Figure 1 was not consistent with its figure legend, since those description is not specific for CRC.

Author Response

We thank the Reviewer for his/her suggestions, which allow us to improve the quality of our manuscript. As requested by the Reviewer, a deeper description of lncRNA molecular mechanisms (please see section 2 “The non-coding RNAs and their deregulation in CRC”) has been included in the revised version of the manuscript. We also improved figure 1 and legend accordingly to new descriptions of molecular mechanisms. Moreover, we modified introduction of section 3 “LncRNAs in initation and progression of CRC” and section 4 “Role of lncRNAs in CRC drug resistance” to clarify the argumets reviewed in those sections.

The manuscript has been now extensively edited and reviewed by Editage, a professional English language editing and publication support service. The certificate of editing is attached.

Reviewer 3 Report

The authors did a good job on summarizing the literature regarding the role of non-coding RNAs in colorectal cancer with a focus on long non-coding RNAs. The topic reviewed is significant and the article will have a good body of audience. This reviewer suggest publication with minor revision. The authors should check the following to see if any changes/improvements should be made (use the Find function). If no change for any of them, it is OK. “Colorectal cancer (CRC) is one of the third most frequently and lethal diagnosed type of cancers worldwide”, “targeted agents adminis-tered based on knowledge of a few tumor biomarkers”, “evolutionary conserved, as a single-stranded RNAs [8].” “a series of up- or down-regulated miRNAs able to predict the response to neoadjuvant ther-apy have been described”, “by circRNAs in in CRC patients with lung metastasis”, “invasion of CRC cells vis activating Hippo-Yap signaling pathway”, “are defined as transcript longer than 200 nucleotides”, “Drugs Substrate”, “Porphyrias [95]. Many others siRNA-based drugs for”, “on new generation antisense oligonucleotide (ASOs)”, “lncRNAs. LANs are”, “by specific LANs in human endothelial”, “be a new therapeutic approach in cancers treatment. In addition”, “role of lncRNAs and their deregulation in CRC as predictive biomarkers”, “some of these lncRNAs may advantage the biomarker-oriented precision medicine approaches”, “provides the basis to for further development of medical applications of lncRNA”, “Alt-hough a wide-ranging analysis and integrated”.

Author Response

We thank the Reviewer for his/her suggestions, which allow us to improve the quality of our manuscript. The manuscript has been now extensively edited and reviewed by Editege, a professional English language editing and publication support service. The certificate of editing is attached.

In addition to language editing, we included in the revised version of the manuscript a deeper description of lncRNAs molecular mechanisms, and a new introduction of section 3 “LncRNAs in initation and progression of CRC” and section 4 “Role of lncRNAs in CRC drug resistance”.  

Round 2

Reviewer 1 Report

The authors addressed my concerns. Thus, I would recommend publishing it.